# Stability analysis of Strange Attractors using Attractor Networks

## Abstract

Understanding the behavior of nonlinear differential equations is an extremely difficult problem. This problem is compounded by the frequent chaotic behavior demonstrated by high-dimensional dynamical systems. A subset of these systems, namely strange attractors, are of particular interest due to their sensitive dependence on initial conditions. Analytical reports on the stability of these attractors rely heavily on the ability to solve the underlying equations. In this work, an attractor network is used for the identification of different regions and the parameters resulting in that particular region, in well-defined strange attractors. The network takes in the initial configuration of the system, stores the pattern of neuronal firing as a state vector, and predicts the behavior of the attractor.

## 1   Introduction

Nonlinear dynamical systems are notoriously difficult to solve analytically. Existing methods like Euler's method Euler (1989) and Runge-Kutta method Runga Kutta (1895) prove to be ineffective when the approximations are set dynamically. Consider solving Equation 1 using Euler's method with an initial condition $y(0) = 0.3$.

$$y' = e^x sin(y) \tag{1}$$

Since this method is based on approximations, any slight deviations in the system's rudimentary phase will have cascaded results. This dependence on initial conditions is a characteristic of strange chaotic attractors David Ruelle (1989). These attractors can display a rich variety of nonlinearities, which usually contain irregular and unpredictable time evolution of globally deterministic systems with a nonlinear coupling of its local variables. However, in certain cases identifying the nature of these nonlinearities is beneficial, as it decides the chaotic regime the system will evolve towards. Events like boundary crisis Grebogi (2016), which is a catastrophic bifurcation in a system, force the system to converge towards a non-chaotic state.

Accomplishing this task often requires a very high level of approximation which is achieved most commonly by universal approximators like neural networks and support vector machines. Recently, the field of deep learning has shown a lot of promise in solving differential equations Serrano-Perez (2021). Adaption of the network parameters provides shows high potential in the manipulation of these attractors and the associated high-dimensional nonlinear systems. One particular architecture of neural networks, namely attractor networks Amit (1992) Han (2018), can retain and forget information regarding the training phase, providing more flexibility in estimating the evolution of a system.

## 2 Methodology

### 2.1 Dynamical System

In our work, we consider dynamical systems which describe the position of all points in a space $S \in \mathbb{R}^n$. Furthermore, the dynamical system is a continuously differentiable function $\phi : \mathbb{R} \times \mathbb{R}^n \to \mathbb{R}^n$. Let $S$ is governed by a system of differential equations $F(X) \in \mathbb{R}^n$. A set $\triangle$, with flow $\phi_t$ is called an attractor if it satisfies the following conditions:

- $\triangle$ is compact and invariant

- $\triangle$ follows the transitive property

- There exists at least one open set $U$ containing $\triangle$ such that for each $X \in U$, $\phi_t(X) \in U$ and $\cap_{t>0} \overline{\phi_t(U)} = \triangle$

These attractors are considered to be *strange* attractors when they display sensitive dependence on initial conditions. Furthermore, the equations governing these attractors depict a bounded region of the phase-space having a positive Lyapunov exponent.

Determining the trajectory of most dynamical systems is difficult since the systems are known to be approximated i.e, the parameters may not be known. These approximations cast doubt on the validity of numerical solutions. One method of addressing these queries is by performing stability analysis tests such as *Lyapunov* stability test Shevitz (1994).

### 2.2 Lyapunov Stability

A point of equilibrium for an attractor is considered to be stable if solutions remain in the neighbourhood for all future timesteps. For a set of differential equations $X' = F(X)$, $X*$ is said to be a stable equilibrium point if, for every neighbourhood $\mathcal{O}$ of $X*$, there is neighbourhood $\mathcal{O}_1$ of $X*$ in $\mathcal{O}$ such that every solution in $X(t)$ with $X(0) = X_0$ in $\mathcal{O}_1$ is defined and remains in $\mathcal{O}$, $\forall t > 0$.

We denote the generated flow by $X \in \mathcal{X}^1(M)$ and let subset $A \subset M$ and subset $R \subset \mathcal{R}$. We define set $X_R(A) = \{X_t(q) : (q,t) \in A \times R\}$. $A \subset M$ is *Lyapunov* stable for $X$ if for every open set $U$ containing $A$ there exists an open set $V$ containing $A$ such that $X_t(V) \subset U \ \forall t > 0$. The key properties of a Lyapunov set $\triangle$ are listed below.

- If $x_n \in M$ and $t_n > 0$ satisfy $x_n \to x \in \triangle$ and $X_{t_n}(x_n) \to y$, then $y \in \triangle$.

- The globally unstable manifold $W(\triangle) \subset \triangle$.

- For a transitive set $\tau$ of $X$ and $\tau \cap \triangle \neq$ , then $\tau \subset \triangle$.

### 2.3 Attractor Networks

Attractor networks are a special class of recurrent neural networks, which have an associative memory component with synaptic feedback loopsYanan (2020). The network takes in the initial configuration of the differential equations and provides the dynamic attractor as an output. These networks store and utilize information regarding the neuronal firing patterns observed during the training phase. In our work, we use a recurrently connected continuous attractor network with $N$ layers evaluated for $t_n$ time intervals. The subset of the network is shown in Figure 1.

The pattern observed by each layer is stored as a state vector. The synapses of the networks then construct the pattern vectors, which correspond to trajectories in the phase space. In this process, the system evolves autonomously allowing for continuous modifications. Note that the layers need to be able to display ergodicity breaking i.e, the state vectors will be confined to a restricted space region, decided by the initial configurations. The continuous attractor network algorithm for manifold prediction is shown in Algorithm 1.

**Algorithm 1** Continous Attractor Network

---

Input: number of layers, number of neurons per layer, time steps for prediction
**for** number_of_layers **do**
   **for** number_of_neurons **do**
      **for** number_of_time_steps **do**
         Initialize model with given parameters
         Train neural network with initialized variable
         Construct phase space diagram and evaluate the trajectories
      **end for**
   **end for**
**end for**

---

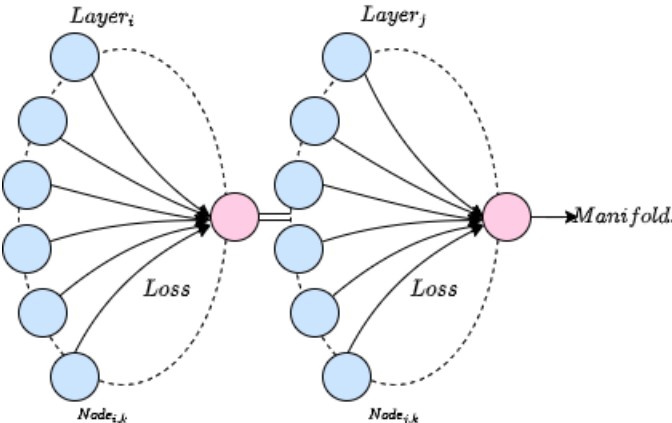

Figure 1: Illustration of a sample of the continous attractor network with two networks. For $N$ layers, the total computational layers are $(N+1)(n-1)$ which invovle free parameters to optimize and predict the manifolds of the equations.

## 3 Experimental Analysis

### 3.1 Strange Attractors

The strange attractor considered in this study are the Lorenz Tucker (1999) and Rossler attractor Rossler (2017). The differential equations defining the Lorenz Attractor and Rossler are given in Equations 2 and 3, respectively.

$$
\begin{aligned}
x' &= \sigma(y - x) \\
y' &= x - (\rho - z) - y \\
z' &= xy - \beta z
\end{aligned}
\tag{2}
$$

$$
\begin{aligned}
x' &= (y - x) \\
y' &= x + ay \\
z' &= b + z(x + c)
\end{aligned}
\tag{3}
$$

### 3.2 Experimental Results

Since both the attractors have an equal number of input and output parameters, the input and output of the network remain the same. The number of layers and the number of nodes per layer are flexible enough to accommodate any outliers. Each network accepts the state vector provided by the previous network, while the training phase is active only for $T_{\text{train}}$ time steps. The exact configurations of the network are mentioned in Table 1. The phase spaces for each attractor with variations in the parameters are shown in Figure 2. The training phase is represented by a different tinge as compared to the prediction phase.

Table 1: Model Configuration for Lorenz and Rossler experiments

| Parameter | Value |
|---|---|
| Number of layers | 10 |
| Neurons per layer | [32,64,128] |
| Training Epochs | 10 |
| Optimizer | Adam |
| Training phase end time | 25 |

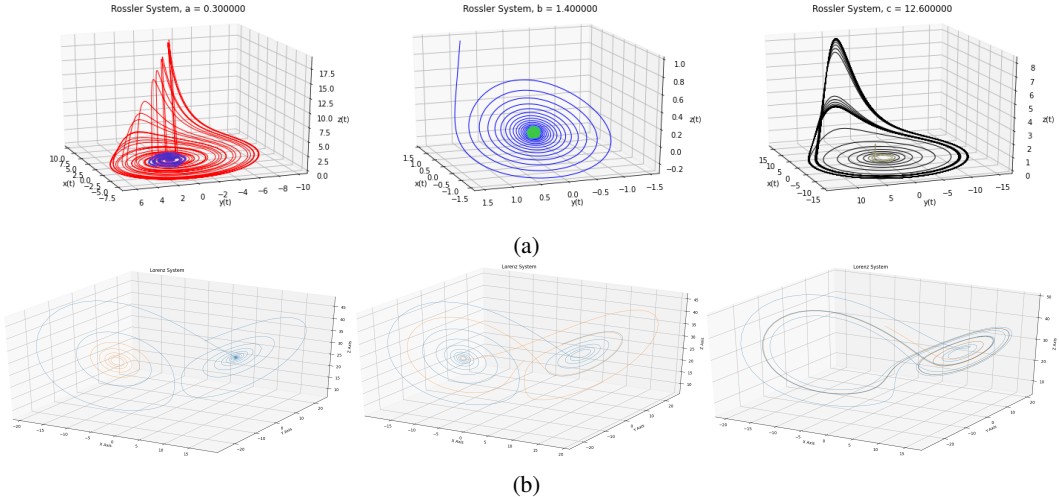

(a)

(b)

Figure 2: Manifolds for (a) Rossler System with parameter set $[a, b, c] =$ $[0.3, 0.2, 5.7], [0.2, 1.4, 5.7], [0.1, 0.1, 12.6]]$, (b) Lorenz System with parameter set $[\sigma, \beta, \rho] = [[10, 8/3, 28], [10, 8/3, 27], [10, 3, 28]]$

## 3.3 Analysis

The experiments were conducted by varying the parameters and the followed behavior was observed. We observed that for the Rossler equations, the system falls into a chaotic state under the following two conditions: $a$ increases beyond 0.2, keeping the other variables constant, and when $c$ is increased beyond 6, keeping the other variables constant. For the Lorenz equations, the system displays chaotic behaviour for $[\sigma, \beta, \rho] = [10, \frac{8}{3}, 27]$. The system reverts to a periodic motion for $\rho$ values greater than 28.

## 4 Conclusion

In this work, we present the applicability of continuous attractor networks to determine the parameter set corresponding to the various stability regions of strange attractors. Two prominent strange attractors, Lorenz and Rossler were considered in this work. The continuous attractor network made predictions based on the initial conditions supplied to the network.

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
