# OpenReview forum: "Stability analysis of Strange Attractors using Attractor Networks"
_NeurIPS.cc/2021/Workshop/DLDE — Reject_

### Official Review · Reviewer_YzTX · 2021-09-30
**Interesting problem, but the presentation needs work**

**Confidence:** 4

**Review:**

Pros:

- The paper tackles an interesting and important problem

Cons:

- It is not easy to understand the specific contributions the authors make
- There are several issues with the presentation (details described below)
- The experiments and corresponding analysis are limited and lack any comparison with alternate methods and so it is difficult to evaluate the contribution from the continuous attractor networks.

Some Details:

General - can the equations be made in the same font size as the text? The align environment should do this automatically. Also, many author name citations are included directly in the text e.g L13, can the authors please use citep to put them in parantheses.
L27 - provides shows
L35 - Let S is governed -> Let S be governed
L47 -  as Lyapunov stability test  -> as the Lyapunov stability test
L62 - Is the Yanan ref. correct? It doesn't mention attractor networks or synapses
Algorithm1 continuous is misspelt.

**Score:**

2: Borderline paper

---

### Official Review · Reviewer_S31p · 2021-10-01
**The paper's contribution is unclear**

**Confidence:** 4

**Review:**

The authors propose the use of attractor networks (a special class of recurrent neural nets) to predict stability regions of strange attractors,
applying the approach to Lorenz and Rossler systems.

Pros:
- It addresses an important problem in the realm of chaotic systems

Cons:
- The main contribution of the paper is unclear: how does it position itself w.r.t. other methods in the literature ? Can these networks outperform in some sense state-of-the-art ways of determining the stability regions ?
- The mathematically detailed descriptions in Sections 2.1 and 2.2 don't help in analyzing the experiments. Some could have been left out for the benefit of a better description of the network's architecture and deeper exploration of its capabilities.
- The training process merits further explanations: which is the "initial configuration of the differential equations" (L62) taken as input ? Algorithm 1 states the model hyperparameters (number of layers, neurons per layer, ...) as input, which is clearly unrelated to the differential equations themselves. Which data correspond to your training set precisely ?
- Fig. 2 labels font size is too small. Fig. 2(b) trajectory drawings are barely visible.

**Score:**

2: Borderline paper

---

### Official Review · Reviewer_DDMX · 2021-10-11
**Interesting framework for solving a difficult problem but hard to see a novel contribution**

**Confidence:** 3

**Review:**

The authors evaluate Deep Attractor Networks on two well known dynamical systems.
Deep Attractor Networks provide an interesting way of encoding information about stability regions of strange attractors in chaotic systems.

Pros:
The framework being proposed is interesting and there should be an opportunity to make a unique contribution to this area of work.

Cons:
The analysis is lacking in discussion of previous works. Mainly, it lacks any discussion about what is the benefit of this method compared to other existing methods.

It lacks discussion about the limitations of Deep Attractor Networks to solve these problems.

The work lacks discussion and mathematical analysis about why Deep Attractor Networks might be particularly good for solving these problems as opposed to some other method.

It's hard to quantitatively judge based on this paper how well the method works to solve the problem.

Some added details about the experiments in place of the details about the underlying theory would be helpful to distinguish any benefits or limitations of Deep Attractor Networks.

Minor issues:

The font size of the equations should be the same as that of the text, it's hard to read the equations with such a small font.

There's a spelling mistake in the caption for figure 1.. Furthermore I believe the diagram shows a sample of two layers in a deep attractor network? Not two networks?

In the experiment section the lower figures should also be colored - I don't see a reason for keeping the color grey, it's hard to see well what's going on.

**Score:**

1: Reject: trivial or wrong

---

### Decision · Program_Chairs · 2021-10-17

**Decision:**

Reject

**Comment:**

While this paper considers an interesting problem at the intersection of chaos theory and deep learning, it is currently missing fundamental elements to be complete enough for a NeurIPS workshop. The authors are encouraged to continue their work while taking reviewer's comments into consideration for future publications.